# Effects of Vitamin A on Yanbian Yellow Cattle and Their Preadipocytes by Activating AKT/mTOR Signaling Pathway and Intestinal Microflora

**DOI:** 10.3390/ani12121477

**Published:** 2022-06-07

**Authors:** Xinxin Zhang, Hongyan Xu, Congcong Zhang, Jinhui Bai, Jixuan Song, Beibei Hao, Luomeng Zhang, Guangjun Xia

**Affiliations:** 1College of Agriculture, Yanbian University, Yanji 133002, China; z329587741@163.com (X.Z.); xuhongyan@ybu.edu.cn (H.X.); cczhang0204@163.com (C.Z.); ybu.bjh@outlook.com (J.B.); 18643612225@163.com (J.S.); haobeibei2021@126.com (B.H.); zlmzka@126.com (L.Z.); 2College of Integration Science, Yanbian University, Yanji 133002, China; 3Engineering Research Center of North-East Cold Region Beef Cattle Science & Technology Innovation, Ministry of Education, Yanbian University, Yanji 133002, China

**Keywords:** vitamin A, adipocyte differentiation, Yanbian yellow cattle, intestinal microflora

## Abstract

**Simple Summary:**

Vitamin A is a fat-soluble vitamin that not only plays a role in vision, growth, and development, but also in fat production and metabolism in animals. To improve the production of high-grade beef, it is necessary to explore the molecular mechanism of intramuscular fat deposition in beef cattle through molecular biology techniques. In this study, we selected Yanbian yellow cattle, one of the five major cattle breeds in China, to investigate the effects of vitamin A and its metabolite, all-trans retinoic acid (ATRA), on the proliferation and differentiation of preadipocytes and changes in intestinal microorganisms. It was found that ATRA inhibited adipogenic differentiation of preadipocytes in Yanbian yellow cattle via the AKT/mTOR signaling pathway. This study provides insight into nutritional management and reveals the role of vitamin A in lipid metabolism in Yanbian yellow cattle.

**Abstract:**

In this study, the effects of vitamin A and its metabolite, all-trans retinoic acid (ATRA), on the proliferation and differentiation of preadipocytes and the intestinal microbiome in Yanbian yellow cattle were investigated. Preadipocytes collected from Yanbian yellow cattle treated with different concentrations of ATRA remained in the G1/G0 phase, as determined by flow cytometry. Quantitative reverse-transcription polymerase chain reaction and western blotting analyses showed that the mRNA and protein expression levels of key adipogenic factors, peroxisome proliferator- activated receptor gamma (PPARγ), CCAAT enhancer-binding protein α (C/EBPα), and extracellular signal-regulated kinase 2 (ERK2), decreased. ATRA was found to regulate the mTOR signaling pathway, which is involved in lipid metabolism, by inhibiting the expression of AKT2 and the adipogenic transcription factors SREBP1, ACC, and FAS; the protein and mRNA expression levels showed consistent trends. In addition, 16S rRNA sequencing results showed that a low concentration of vitamin A promoted the growth of intestinal microflora beneficial to lipid metabolism and maintained intestinal health. The results indicated that ATRA inhibited the adipogenic differentiation of preadipocytes from Yanbian yellow cattle through the AKT/mTOR signaling pathway, and that low concentrations of vitamin A may help maintain the intestinal microbes involved in lipid metabolism in cattle.

## 1. Introduction

Vitamin A, also known as retinoic acid, is synthesized from β-carotene in the small intestine and liver of animals [1]. Ruminants typically obtain vitamin A from plant sources and feed additives [2]. Vitamin A was found to be negatively associated with marbling in beef [3]; a 16% higher marbling score was observed in Angus cull cattle with normal levels of vitamin A compared to those continuously fed low levels of vitamin A [4]. This demonstrated that vitamin A plays an important role in fat accumulation in beef cattle, and that its metabolite, all-trans retinoic acid (ATRA), is a regulator of early adipocyte differentiation and lipid metabolism [5]. Previous studies have found that ATRA mainly inhibits fat deposition in the body by reducing the proliferation and differentiation of preadipocytes, inducing apoptosis, and promoting the degradation of adipocytes [6]. 

The Yanbian breed of yellow cattle is one of the five most abundant breeds of cattle in China [7]. This breed has the advantages of cold tolerance, coarse feeding tolerance, and good meat production. Meat quality traits, such as lean meat percentage and intramuscular fat deposition, are directly affected by the amount of adipose tissue, which affects tenderness and flavor; these aspects are important in animals raised for meat. Thus, controlling total fat deposition and distribution will help improve the availability of animals raised for meat [8]. The formation of adipose tissue in beef cattle follows three stages: fat formation, fat differentiation, and lipid accumulation [9]. Adipose tissue is mainly composed of fat cells and blood vessels, and participates in energy metabolism and maintaining the shape of tissues and organs [10].

In the process of differentiation from preadipocytes to mature adipocytes, adipogenic transcription factors, such as peroxisome proliferator-activated receptors (PPARs), CCAAT enhancer-binding protein, and sterol regulatory-element binding proteins (SREBPs) play important roles [11]. PPARs and CCAAT enhancer-binding protein are involved in the early stage of adipocyte differentiation and can be detected before differentiation. As differentiation progresses, gene expression levels increase rapidly, reaching a peak when the preadipocytes differentiate into mature adipocytes [12]. SREBPs are key nuclear transcription factors that are closely related to fatty acid synthesis, cholesterol metabolism, and fat differentiation. They are also key regulators of the mTOR pathway, which plays an important role in regulating lipid metabolism in adipocytes. The mTOR pathway senses changes in insulin levels in animals and regulates lipid production by affecting insulin signaling [13].

In addition, some studies have shown that intestinal microbes regulate fat metabolism and affect the level of fat deposition in tissues [14]. Intestinal microbes directly regulate the expression levels of host fat-storage genes, promote the accumulation of fat in the host [15], and affect the energy balance of the host through changes in their composition or by regulating enzymes related to fat metabolism [16]. The thick-walled phylum and the bacteriophage phylum, as the dominant groups, have a role in promoting the degradation of fibrous material and the digestion and absorption of carbohydrates [17]. Therefore, the purpose of this study was to explore the effects of ATRA on adipogenic differentiation and the mTOR pathway in Yanbian yellow cattle and to determine the effects of vitamin A feeding on the intestinal microflora of the cattle.

## 2. Materials and Methods

### 2.1. Cell Culture and Adipogenic Differentiation

Preadipocytes were obtained from Yanbian yellow cattle from the Jilin Hunchun Ji Xing Herding Co. Ltd. (Jilin, China). The cells were cultured in complete Dulbecco’s modified Eagle’s medium (Gibco, Waltham, MA, USA) with 10% fetal bovine serum (FBS) at 37 °C and 5% CO_2_. Contact inhibition was achieved when 100% confluence occurred. At this time, the cells were induced to differentiate, and this was recorded as day 0 of differentiation. The initial differentiation medium was complete medium containing 1 μmol/L dexamethasone (Sigma-Aldrich, St Louis, MO, USA); 1 mg/L insulin; 500 μmol/L 3-isobutyl-1-methylxanthine; and 2 × 10^−5^, 1 × 10^−5^, 2 × 10^−6^, 1 × 10^−6^, or 2 × 10^−7^ mol/L ATRA. The ATRA was purchased from Nanjing Bai Xin de Nuo Biotechnology Co. Ltd. (Nanjing, China). On the third day of differentiation, the medium was changed to a complete medium containing 10 μg/mL insulin. On the fifth day of differentiation, the medium was replaced again with a complete medium containing 10% FBS and the same ATRA concentration. The medium was changed every two days until the ninth day when the differentiation process was completed.

### 2.2. Measurement of Cell Proliferation

3-(4,5-Dimethylthiazol-2-yl)-5-(3-carboxymethoxyphenyl)-2-(4-sulfophenyl)-2H-tetrazolium (MTS) staining solution was purchased from Promega (Madison, WI, USA), and cell viability was evaluated according to the manufacturer’s instructions. Cells were seeded in 96-well plates at a density of 1 × 10^4^ cells/mL and cultured for 12 h at 37 °C in a constant temperature incubator with 5% CO_2_. Then, ATRA was added at various concentrations (2 × 10^−5^, 1 × 10^−5^, 2 × 10^−6^, 1 × 10^−6^, or 2 × 10^−7^ mol/L). MTS solution (20 μL) was added on days 0, 1, 2, 4, 6, and 8, and, after culturing the cells in a CO_2_ incubator for 2 h, the optical density of each well was measured using a Spark multifunctional microplate reader (Tecan, Männedorf, Switzerland).

### 2.3. Cell Cycle Analysis

Cells were inoculated into six-well plates and ATRA (2 × 10^−5^, 1 × 10^−5^, 2 × 10^−6^, 1 × 10^−6^, or 2 × 10^−7^ mol/L) was added after 24 h. When the cells reached 50–80% confluence, they were washed with phosphate-buffered saline (PBS) and trypsinized in the original culture medium. After mixing, the digested cells were centrifuged at 1000 rpm for 5 min. The resulting supernatant was discarded, 1.5 mL of precooled PBS was added, and 3.5 mL of anhydrous ethanol was added while vortexing. The cells were mixed evenly and fixed at 4 °C for 30 min, after which they were centrifuged at 1000 rpm for 5 min. The ethanol was removed, and the cells were washed with PBS, mixed, and centrifuged again. Twenty microliters of PBS and 2 μL of RNase (0.25 mg/mL) were added, and the cells were incubated at 37 °C for 30 min. The cells were then stained with 0.5 mL of propidium iodide solution at 4 °C for 30 min. After filtration through a 300-mesh nylon membrane, the cell cycle stage was determined by flow cytometry.

### 2.4. Oil Red O Staining

On the ninth day of differentiation, the cells were washed twice with PBS and fixed with 10% formaldehyde for 30 min at room temperature. After washing twice with PBS, the cells were stained for 40 min with Oil Red O (Sigma-Aldrich, St. Louis, MO, USA), dissolved in 60% isopropyl alcohol, and evenly mixed with double-distilled water at a volume ratio of 6:4. Stained lipids were observed under an optical microscope (Axio Imager; Zeiss, Oberkochen, Germany). The Oil Red O dye was extracted with isopropanol, and the absorbance was measured at 520 nm using a microplate reader.

### 2.5. RNA Extraction and Quantitative Polymerase Chain Reaction Analysis

Total RNA was extracted from cells using an Eastep Super Total RNA extraction kit (Promega, Beijing, China). The RNA was reverse-transcribed into cDNA in two steps using a PrimeScript Reagent Kit with gDNA Eraser (Perfect Real Time; Takara, Shiga, Japan). Real-time polymerase chain reaction (PCR) was performed using a fluorescent quantitative kit TB Green™ Premix Ex Taq™ II (Dalian Bao Bio-Company, Dalian, China) and a PCRmax Eco 48 real-time PCR instrument. Each sample was assayed in triplicate. The relative expression of each gene at the mRNA level was calculated using the 2^−ΔΔCT^ method. β-actin was used as a housekeeping gene to normalize the expression levels of the other protein-coding genes. The sequences of the PCR primers are shown in Table 1.

### 2.6. Protein Extraction and Western Blotting Analysis

Radioimmunoprecipitation assay buffer lysate (Beyotime, Shanghai, China), 100 mM phenylmethylsulfonyl fluoride (Beyotime), and phosphatase inhibitor were mixed in a ratio of 100:1:1 and used to lyse cells on ice. After obtaining total cellular protein, an enhanced bicinchoninic acid protein detection kit (Beyotime) was used to measure protein concentration according to the manufacturer’s instructions. Proteins were separated by sodium dodecyl sulfate polyacrylamide gel electrophoresis and transferred to polyvinylidene fluoride membranes according to the manufacturer’s instructions. After blocking with 5% bovine serum albumin at room temperature for 2 h, the membranes were washed five times with a Tris-buffered saline-Tween 20 (TBST) solution. Membranes were incubated with the primary antibody overnight at 4 °C and then washed with TBST for 30 min. After incubating the membranes with the secondary antibody for 2 h at room temperature, they were washed with TBST for 30 min. The membranes were exposed using the Alliance MINI HD9 AUTO protein Imprint Imaging System (American Ultraviolet Technology Company, Lebanon, IN, USA). The protein bands were quantified using ImageJ software (National Institutes of Health, Bethesda, MD, USA).

### 2.7. Animals and Feeding

During this study, each animal was equipped with an ear tag to track its progress and growth. Fifteen healthy Yanbian yellow cattle with similar body weight (initial 330.5 ± 15.9 kg; approximately 15 months old) were divided into five groups, each with three head of cattle. Prior to the trial, we fed the cattle for 15 days according to the trial design as an adaptation period. We began formal testing once there were no anomalies. For the control group, the formal trial diet was supplemented with 2200 IU vitamin A/kg dry matter (DM), according to the recommendations for beef cattle in the growth and fattening stages in the National Research Council publication, Nutritional Requirements of Beef Cattle (Eighth Revision). The remaining 12 cattle were divided into the following experimental groups: the NVA1 group, fed 0 IU of vitamin A/Kg DM for 180 days; the NVA2 group, fed 0 IU of vitamin A/Kg DM for 240 days; the LVA1 group, fed 1100 IU of vitamin A/Kg DM for 180 days; and the LVA2 group fed 1100 IU vitamin A/Kg DM for 240 days. Vitamin A was purchased from the DSM Vitamin Co., Ltd. (Changchun, China). During the study, all cattle were fed a standard basic diet and were free to drink water. The feed concentrate was controlled at 1% of the body weights of each cattle and adjusted once a month according to the body weights of the cattle. The composition and nutritional components of the basic diet are presented in Table 2.

### 2.8. Diversity Analysis of Intestinal Flora of Yanbian Yellow Cattle

Feces were collected in a sterile environment, immediately frozen in liquid nitrogen to extract DNA, and stored at −80 °C until use. Total DNA of the intestinal microflora was extracted from stool samples using a bacterial DNA extraction kit (BioTeke, Beijing, China), according to the manufacturer’s instructions. To construct a sequencing library, the V3–V4 region of the 16S rRNA gene was amplified in a reaction containing 5 μL of 5× reaction buffer, 5 μL of 5× GC buffer, 2 μL of dNTPs (2.5 mm), 1 μL of forward primer (10 M), 1 μL of reverse primer (10 M), 2 μL of DNA template, 8.75 μL of double-distilled H_2_O, and 0.25 μL of Q5 high-fidelity DNA polymerase (New England Biolabs, Ipswich, MA, USA). The amplification parameters included: an initial denaturation at 98 °C or 2 min; 25–30 cycles of 98 °C for 15 s, 55 °C for 30 s, and 72 °C for 30 s; and a final extension at 72 °C for 5 min. PCR products were detected and purified by 2% agarose gel electrophoresis, and the target fragments were recovered using a gel recovery kit (Axygen, Union City, CA, USA). The amplification products were mixed in equimolar ratios and quantified using a Quantum dsDNA detection kit. Finally, a DNA sequencing library was prepared using a TruSeq Nano DNA LT library preparation kit (Illumina, San Diego, CA, USA), verified using an Agilent high-sensitivity DNA kit (Agilent, Santa Clara, CA, USA), and then used for sequencing on an Illumina MiSeq platform.

### 2.9. Statistical Analysis

All results are expressed as the mean ± standard deviation. Quantitative PCR data were analyzed using an independent-sample Student’s *t*-test, with *p* < 0.05 indicating a statistically significant difference between groups. All statistical analyses were performed using SPSS 26 software (IBM, Armonk, NY, USA).

## 3. Results

### 3.1. Effect of ATRA on the Proliferation of Preadipocytes from Yanbian Yellow Cattle

The MTS method and flow cytometry were used to determine the effects of ATRA on cytotoxicity and cell cycle progression. Our results showed that, compared with the control group, the optical density values of each ATRA treatment group showed no significant change on the first day, which indicated that the selected concentrations did not affect preadipocyte viability and could be used in subsequent experiments (Figure 1a). Subsequently, flow cytometry analysis indicated that ATRA treatment resulted in cell cycle arrest at the G1/G0 phase, resulting in a significant decrease in the number of cells in the G2 phase (Figure 1b). In addition, we found that the expression levels of the cell proliferation-related genes, proliferating cell nuclear antigen (PCNA), cyclin-dependent kinases2 (CDK2), and cyclin D1, were significantly reduced after ATRA treatment (*p* < 0.01; Figure 1c). These results indicated that ATRA inhibited the proliferation of preadipocytes from Yanbian yellow cattle.

### 3.2. Effect of ATRA on the Differentiation of Preadipocytes from Yanbian Yellow Cattle

Oil Red O was used to stain cells on the ninth day after differentiation. With increasing ATRA concentrations, the number of lipid droplets in the cells gradually decreased (Figure 2a). The optical density values of each ATRA treatment group were significantly lower than those of the control group, and the inhibitory effect was more evident at higher concentrations (Figure 2b). These results showed that ATRA significantly inhibited lipid accumulation in preadipocytes from Yanbian yellow cattle (*p* < 0.01) in a concentration-dependent manner.

Quantitative reverse-transcription PCR analysis showed that ATRA downregulated the expression levels of lipid-related genes, including peroxisome proliferator-activated receptor gamma (PPARγ), CCAAT enhancer-binding protein α (C/EBPα), and extracellular signal-regulated kinase 2 (ERK2), whereas it had a contrasting effect on adiponectin (ADPN) (Figure 2c). The mRNA expression levels of PPARγ, C/EBP, and ERK2 were significantly lower in each ATRA treatment group compared with those in the control group (*p* < 0.01). However, the expression levels of ADPN mRNA in preadipocytes from Yanbian yellow cattle treated with different concentrations of ATRA changed irregularly compared with those in the control group. There was no significant difference between the control group and the 1 × 10^−6^ mol/L ATRA treatment group (*p* > 0.05), but there was a significant difference between the control group and the 1 × 10^−5^ mol/L ATRA treatment group (*p* < 0.05). In addition, there were highly significant differences in each treatment group (*p* < 0.01). However, with increasing ATRA concentration, ADPN gene expression levels showed an upward trend, which indicated that ATRA upregulated the expression of ADPN.

We then detected the expression levels of PPARγ, C/EBPα, ERK2, and ADPN proteins after ATRA treatment (Figure 2d). Compared with their levels in the control group, the expression levels of PPARγ, C/EBPα, and ERK2 proteins decreased by different degrees for different ATRA treatment concentrations. PPARγ protein levels were significantly different between the control group and the 2 × 10^−7^, 2 × 10^−6^, 1 × 10^−6^, and 1 × 10^−5^ mol/L ATRA treatment groups (*p* < 0.05) and were highly significantly different between the control group and other treatment groups (*p* < 0.01). The expression levels of C/EBPα protein were significantly different between the 1 × 10^−6^ and the 2 × 10^−6^ and 1 × 10^−5^ mol/L ATRA treatment groups (*p* < 0.05), whereas there was no significant difference between the 1 × 10^−6^ and 2 × 10^−7^ mol/L ATRA treatment groups (*p* > 0.05). The ERK2 protein expression levels were significantly different between the 1 × 10^−6^ and 2 × 10^−6^ mol/L ATRA treatment groups (*p* < 0.05), but there was no significant difference between the 1 × 10^−5^ and 2 × 10^−5^ mol/L ATRA treatment groups (*p* > 0.05). Each other group showed highly significant differences (*p* < 0.01). The levels of ADPN protein showed no obvious pattern but generally increased after ATRA treatment. ADPN protein expression levels were lowest in the 2 × 10^−7^ mol/L ATRA treatment group and were significantly different from the ADPN levels in the other ATRA treatment groups (*p* < 0.01). There were no significant differences in ADPN levels between the other treatment groups (*p* > 0.05). These results showed that ATRA inhibited the expression of PPARγ, C/EBPα, and ERK2, and that the protein and mRNA levels tended to be consistent.

### 3.3. ATRA Regulated the Differentiation of Preadipocytes from Yanbian Yellow Cattle through the mTOR Signaling Pathway

The mTOR signaling pathway is an effective promoter of lipid synthesis via the insulin-AKT pathway. AKT2, an important regulatory factor, is mainly expressed in the fat, liver, and other tissues. We determined the gene expression levels of the upstream factor of the mTOR pathway, AKT2, and the important regulatory factors of the mTOR pathway, SREBP1, ACC, and FAS (Figure 3a). AKT2 mRNA expression in Yanbian yellow cattle preadipocytes was significantly inhibited by ATRA treatment in a dose-dependent manner (*p* < 0.01). There were significant differences in these mRNA levels between the 2 × 10^−5^, 1 × 10^−5^, and 2 × 10^−6^ mol/L ATRA treatment groups (*p* < 0.05); no significant differences between the 1 × 10^−5^ and 2 × 10^−6^ mol/L ATRA treatment groups (*p* > 0.05); and highly significant differences among the other treatment groups (*p* > 0.05). Thus, ATRA inhibited AKT2 expression, resulting in the regulation of the mTOR pathway downstream of ATRA. SREBP1 mRNA expression levels showed no significant difference between the 2 × 10^−6^ and 1 × 10^−5^ mol/L ATRA treatment groups (*p* > 0.05), but there were significant differences among the other groups (*p* < 0.01). As the ATRA concentration increased, the expression levels of SREBP1 showed a linear downward trend. ACC mRNA expression levels showed a downward trend in all treatment groups, although there was no obvious pattern. The inhibitory effect on ACC mRNA expression levels was most obvious in the 2 × 10^−5^ mol/L ATRA treatment group, and there were significant differences between the 2 × 10^−6^ and the 1 × 10^−6^, 1 × 10^−5^, and 2 × 10^−5^ mol/L ATRA treatment groups (*p* < 0.05). There were also significant differences in ACC mRNA expression levels between the 1 × 10^−5^ and 2 × 10^−5^ mol/L ATRA treatment groups (*p* < 0.05); no significant differences between the 1 × 10^−6^ and 1 × 10^−5^ mol/L ATRA treatment groups (*p* > 0.05); and highly significant differences among the other treatment groups (*p* < 0.01). The expression levels of FAS mRNA decreased significantly in all treatment groups. There was no significant difference in FAS mRNA levels between the 1 × 10^−5^ and 2 × 10^−5^ or the 2 × 10^−7^ and 1 × 10^−6^ mol/L ATRA treatment groups (*p* > 0.05), but there was a significant difference between the 2 × 10^−6^ and 1 × 10^−6^ mol/L ATRA treatment groups.

The AKT2 protein expression levels in preadipocytes from Yanbian yellow cattle treated with ATRA decreased to different degrees (Figure 3b). There was a significant difference in AKT2 protein expression levels between the 2 × 10^−7^ mol/L ATRA treatment group and the control group (*p* < 0.05), but there was no significant difference between the 2 × 10^−5^ and 1 × 10^−5^ mol/L ATRA treatment groups (*p* > 0.05). There were highly significant differences in AKT2 protein expression levels among the other treatment groups (*p* < 0.01). The effects of ATRA on AKT2 protein expression levels were dose-dependent. After ATRA treatment, the expression levels of SREBP1 protein decreased significantly, and there was no significant difference between the 2 × 10^−6^ and 1 × 10^−5^ mol/L ATRA treatment groups (*p* > 0.05) or between the 1 × 10^−6^ and 2 × 10^−7^ mol/L ATRA treatment groups (*p* > 0.01). These results show that ATRA inhibited the expression of AKT2 and SREBP1, which are key factors in the mTOR signaling pathway, thus inhibiting the expression of FAS and ACC downstream of SREBP1 and participating in the regulation of the mTOR signaling pathway. The trend in AKT2 and SREBP1 protein expression levels after ATRA treatment was consistent with the trend in their respective mRNA levels.

### 3.4. Effect of Vitamin A on the Diversity of the Intestinal Flora of Yanbian Yellow Cattle

Alpha diversity is a comprehensive index of the richness and evenness of an ecosystem, and it is a basic component of biodiversity. The curves of the Chao1 and Shannon indices entered a plateau period and reached saturation, which indicated that the number of sequencing samples was sufficient to fulfill the requirements for assessing microbial diversity (Figure 4a,b). Therefore, these results accurately reflect the level of species diversity of microorganisms in the sequencing samples. The main purpose of β diversity analysis is to investigate the similarity of community structures among different samples. Principal component analysis was used to evaluate the intestinal microbial community structure. As shown in Figure 4c, PC1 and PC2 represented the top two principal axes of data interpretation after dimension reduction, which accounted for 79.8% and 11.5% of data interpretation, respectively. The diversity in the NVA2 group was scattered and clearly separated from CON, LVA1, and NVA1 groups, while one sample in the LVA2 group was an outlier. There was little difference in community composition among the CON, LVA1, and NVA1 groups, indicating that the composition of the intestinal microflora community was similar before and after 180 days of vitamin A supplementation, and, thus, was minimally affected by the concentration of vitamin A administered. In addition, the non-metric multidimensional scaling (NMDS) analysis, another type of clustering representation method, showed similar results to those of the principal component analysis (Figure 4d). These results showed that reducing the concentration of vitamin A in the diet may influence the intestinal microbes of Yanbian yellow cattle; however, with time, this influence of vitamin A concentration could decrease or even disappear.

### 3.5. Vitamin A Regulated the Bacterial Composition in Yanbian Yellow Cattle at Different Taxa Levels

To study the specific bacterial changes, the relative bacterial abundance at different taxa levels was quantified in feces samples. We specifically analyzed bacteria related to lipid metabolism. At the phylum level, the relative abundance of *Proteobacteria* increased significantly in the LVA1 group compared with the control group (*p* < 0.05). After reducing the vitamin A concentration, the abundance of Bacteroides showed significant differences. The relative abundance of *Bacteroides* decreased significantly in the NVA1 and LVA2 groups (*p* < 0.01) but increased significantly in the NVA2 group compared with the control group (*p* < 0.05). Although there were no significant differences, the relative abundance of the phylum *Scleroderma* increased by varying degrees in the NVA1, NVA2, and LVA2 groups (Figure 5a).

At the genus level, the relative abundance of *Alistipes* was significantly higher in the LVA1 group than the control group (*p* < 0.05). Compared with the control group, the relative abundance of the genus *Spirillum* showed a downward trend, decreasing significantly in the NVA1 group (*p* < 0.05), while showing a non-significant downward trend in the NVA2, LVA1, and LVA2 groups (Figure 5b). These results indicated that reducing the vitamin A content may affect the intestinal microbial structure in Yanbian yellow cattle and maintain the intestinal microbial community.

## 4. Discussion

Yanbian yellow cattle exhibit good meat production and a unique meat flavor. However, there are large differences in fat deposition among individual animals, and they show unstable meat quality traits. Therefore, the molecular mechanisms of intramuscular fat deposition urgently need to be investigated in Yanbian yellow cattle using molecular biological techniques to improve the quality of the beef and increase the output of high-grade beef. The growth and proliferation of preadipocytes is the basis of intramuscular fat formation and is important for the synthesis of adipose tissue. Therefore, we selected preadipocytes from Yanbian yellow cattle for these experiments. Cytotoxicity was detected using the MTS method, the effects of different ATRA concentrations were determined, and cell cycle progression was analyzed by flow cytometry. To further clarify the mechanism, we determined the mRNA expression levels of PCNA, CDK2, and cyclin D1 genes and found that the expression levels of each gene decreased significantly after ATRA treatment. Therefore, ATRA may inhibit the proliferation of preadipocytes in Yanbian yellow cattle.

Fat is the main site of vitamin A storage and metabolism, and it is also the main target of ATRA [18]. It is generally believed that different concentrations of ATRA have different effects on adipocyte differentiation. The inhibitory effect of ATRA on preadipocyte differentiation was found to be dose-dependent. After treatment with a high concentration of ATRA, almost no lipid droplets were formed in cells [19], and the mRNA expression levels of PPARγ and C/EBPα were highly significantly reduced, which significantly inhibited preadipocyte proliferation and differentiation [20]. In our experiment, we made similar observations. Different concentrations of ATRA inhibited cell proliferation, adipogenic differentiation, and lipid droplet formation. In addition, during the process of preadipocyte differentiation, ATRA significantly decreased the mRNA and protein expression levels of the adipogenic factors PPARγ, C/EBPα, and ERK2. The mRNA and protein expression levels of ADPN showed an upward trend. Overall, these results indicated that ATRA inhibited the differentiation of preadipocytes from Yanbian yellow cattle.

To further clarify the mechanism of the effect of ATRA on Yanbian yellow cattle preadipocyte differentiation, we determined the mRNA and protein expression levels of mTOR pathway-related genes. AKT2, an upstream factor in the mTOR signaling pathway, directly promotes the absorption of glucose and concurrently activates mTORC1 through the AKT-TSC1/2-RheB-mTORC1 pathway, which can regulate the mTOR signaling pathway. Decreasing the expression level of AKT2 has been shown to significantly inhibit adipogenic differentiation and lipid droplet formation in porcine preadipocytes, significantly decrease the number of lipid droplets and fatty acid synthesis, and decrease the expression levels of PPAR, AP2, C/EBP, and other key adipogenic factors [21]. Our results showed that, compared with the control group, the expression levels of AKT2 mRNA and protein decreased significantly after treatment with different concentrations of ATRA (*p* < 0.01), and the effect was most obvious (75%) at a treatment concentration of 2 × 10^−5^ mol/L. This was consistent with the results of lipid droplet aggregation and indicates that ATRA inhibited the expression of AKT2 at the mRNA and protein levels. In addition, the mTOR signaling pathway activates PPAR gene expression and positively regulates the SREBP1 gene, thus regulating animal lipid metabolism. SREBP1 is an important regulator of the mTOR signaling pathway. After activation, SREBP1 promotes the transcriptional activity of FAS and ACC genes, accelerates the generation of the PPARγ ligand to activate PPARγ, significantly increases triglyceride concentration, and improves the fat conversion efficiency in vivo [22,23,24,25]. Previous studies have shown that SREBP1 inhibits the differentiation of preadipocytes [26]. ATRA downregulates the expression level of SREBP1 by regulating the phosphorylation of C/EBPβ, thus inhibiting the differentiation of 3T3-L1 preadipocytes [27]. FAS and ACC are located downstream of SREBP1 in the mTOR signaling pathway. As enzymes involved in de novo fatty acid synthesis, FAS and ACC are regarded as the core factors for long-term regulation of fat production. ATRA was found to significantly decrease the expression level of the FAS gene [19]. The ACC expression levels were significantly lower in preadipocytes from beef cattle treated with retinoic acid than in control adipocytes, and this effect was dose-dependent [28]. Our results showed that, after ATRA treatment, the mRNA and protein expression levels of SREBP1, ACC, and FAS decreased significantly, which was consistent with the results. Therefore, ATRA inhibits the expression levels of the downstream factors, FAS and ACC, by inhibiting SREBP1 expression and regulating the mTOR signaling pathway to inhibit the differentiation of Yanbian yellow cattle preadipocytes.

Previous studies have shown that 50–60% of carotenoids can be used by animals and converted into vitamin A in vivo [29], while β-carotene has the highest vitamin A conversion efficiency of all the carotenoids [30]. The small intestine is the main site of carotenoid absorption and biotransformation. Before vitamin A is absorbed, it must be emulsified in the small intestine, together with fat from the diet, to form small micelles. This process requires bile salts [31]. Bile acid has an important role in lipid metabolism. It regulates the activities of pancreatic lipase and lipoprotein esterase, increases the hydrolysis and metabolism of fat, transports fat in the intestinal tract, and promotes fat absorption [32]. Bile acid, as a signaling molecule that connects the intestine and the liver, also plays an important role in liver and intestine metabolism [33]. It directly inhibits the growth of intestinal microbes, changes the composition of the intestinal flora, and indirectly regulates the structure of the intestinal flora through the bile acid receptor, FXR. The intestinal flora and bile acids do not act in one direction but interact with each other [34]. Therefore, linking the intestinal flora with lipid metabolism through bile acid signaling clarifies the interaction between intestinal microbes and vitamin A. Using 16S rRNA sequencing, we studied the effect of vitamin A on the rectal microbial composition of Yanbian yellow cattle. Our results showed that the intestinal microbial composition was altered in Yanbian yellow cattle fed a low concentration of vitamin A. The bacteria with altered abundance included *Proteus*, *Bacteroides*, *Spirillum*, *Vibrio*, and *Alistipes*. Reportedly, a smaller number of Proteobacteria results in a more normal intestinal tract in mammals [35]. In this study, we found that the number of *Proteobacteria* was significantly lower in the control group than in the LVA2 group, which indicated that the intestinal tracts of the animals in the control group were in a healthy state. The dominant flora in the gastrointestinal tract of ruminants is *Bacteroides*, which mainly function to promote the digestion and absorption of proteins and carbohydrates. A decline in the abundance of *Bacteroides* has been shown to cause obesity in mice [36], which is consistent with our results.

*Spirillum vibroseis* was found to produce short-chain fatty acids, such as butyrate [37]. A low abundance of *Spirillum vibroseis* is highly associated with obesity-related metabolic diseases. A high-fat diet affects the balance of the intestinal flora, specifically increasing the abundance of *Alistipes* and *Bacteroides* [38]. In our study, feeding Yanbian yellow cattle a low concentration of vitamin A resulted in a decrease in the abundance of the genus *Spirillum* in the intestines to varying degrees and an increase in the abundance of the genus *Alistipes*. Therefore, vitamin A may help maintain intestinal microbial colonies in Yanbian yellow cattle. However, owing to differences in the ecological characteristics of intestinal microbes among individuals, these results only provided evidence for the effect of vitamin A on the regulation of lipid and energy metabolism. The relationship between the intestinal microflora and related gene regulation remains unclear, and, thus, further research on the intestinal-liver axis is required. In the future, we may be able to screen for physiological indicators or protein markers, with predictive ability, to detect changes in the microbial community structure in the intestinal tract and thus provide an early indication of fat metabolism or marbling traits.

## 5. Conclusions

In summary, the results of this study indicate that ATRA reduced the proliferation and differentiation of preadipocytes and significantly downregulated the expression levels of SREBP1, ACC, and FAS by inhibiting AKT2, an upstream factor in the mTOR signaling pathway. Although we did not identify the optimal ATRA concentration beneficial for fat metabolism, feed supplemented with a low concentration of vitamin A helped maintain the intestinal microbes associated with fat metabolism in Yanbian yellow cattle. Thus, metabolites produced in the intestine may have inhibited fat metabolism.

## Figures and Tables

**Figure 1 animals-12-01477-f001:**
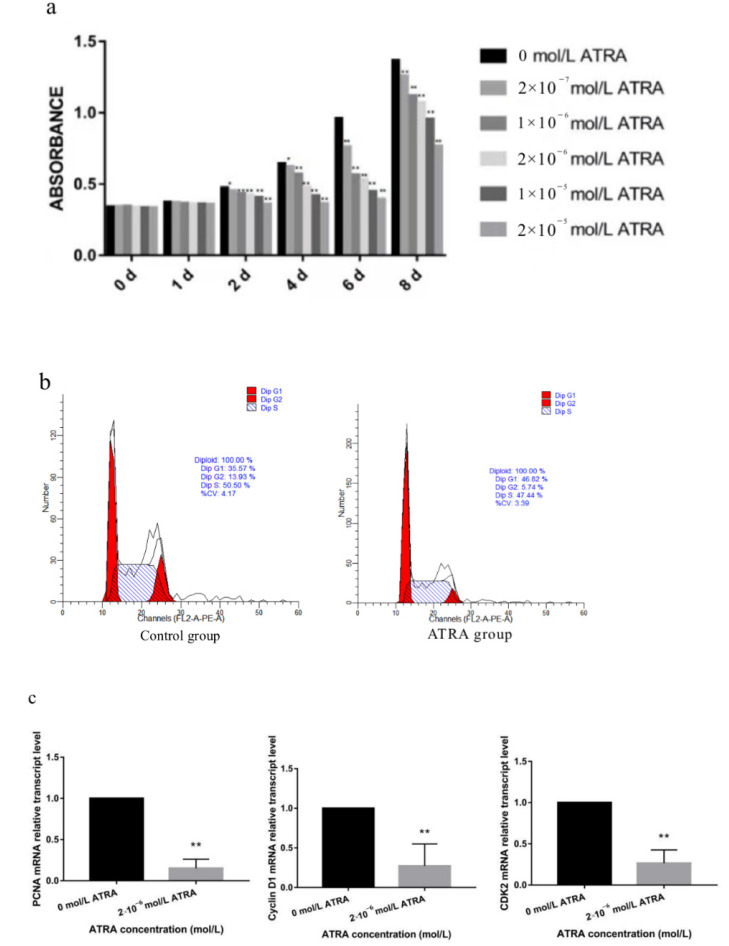
Effect of ATRA on the proliferation of preadipocytes in Yanbian yellow cattle. (**a**) The MTS method was used to determine the effects of different concentrations of ATRA (0, 2 × 10^−7^, 1 × 10^−6^, 2 × 10^−6^, 1 × 10^−5^, 2 × 10^−5^ mol/L) on the viability of preadipocytes from Yanbian yellow cattle. (**b**) Flow cytometry was used to determine the number of cells in different cell cycle stages. mRNA expression levels of (**c**) *PCNA*, cyclin D1, and *CDK2*. * indicates a significant difference compared with the blank control group, *p* < 0.05; ** indicates a highly significant difference compared with the blank control group, *p* < 0.01.

**Figure 2 animals-12-01477-f002:**
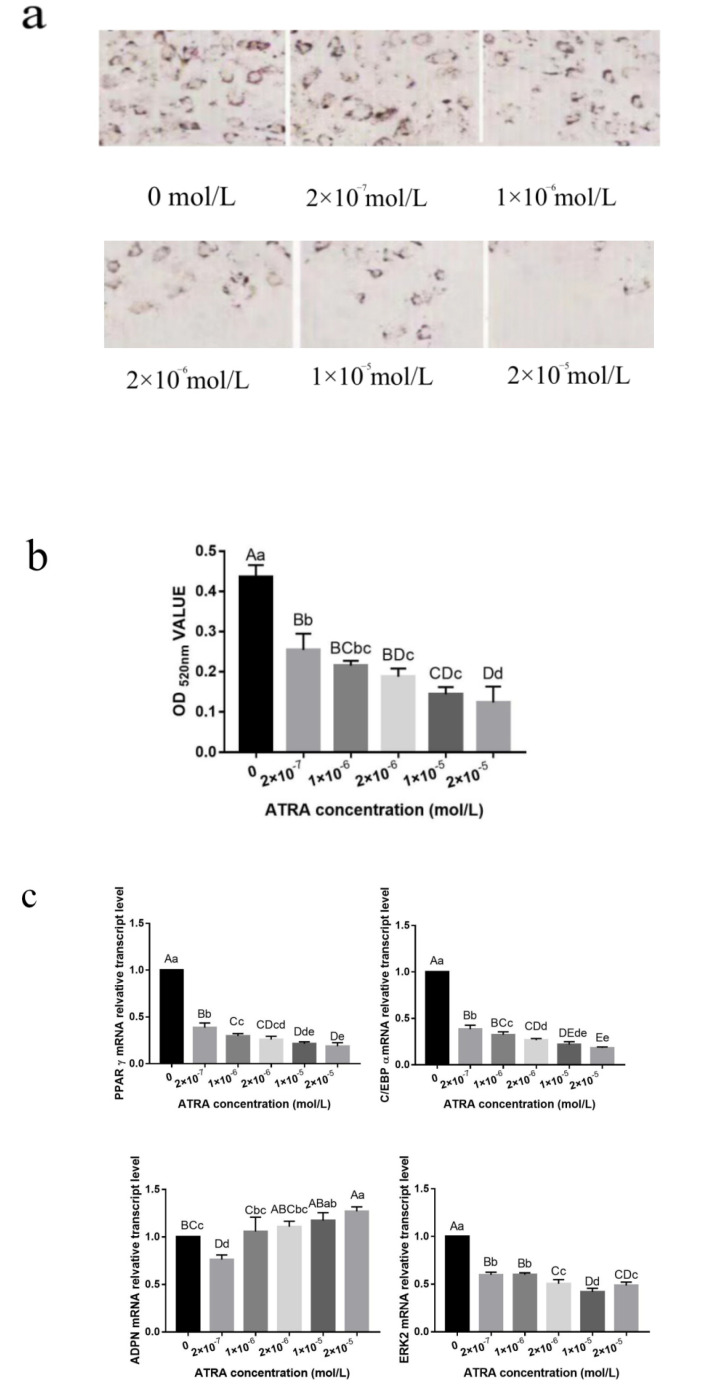
Effect of ATRA on the differentiation of preadipocytes from Yanbian yellow cattle. (**a**) Oil Red O staining of pre-adipocytes (200×) after treatment with different concentrations of ATRA after the ninth day of differentiation. (**b**) The lipid content of differentiated cells was quantified by extracting Oil Red O with isopropanol and measuring its absorption at 520 nm. (**c**,**d**) Quantitative reverse-transcription polymerase chain reaction (qRT-PCR) and western blotting were used to determine the levels of PPARγ, C/EBPα, ERK2, ADPN (**c**) mRNA and (**d**) protein in ATRA treatment groups. Different lowercase letters in the figure indicate significant differences compared with the control group, *p* < 0.05. Different capital letters indicate highly significant differences compared with the control group, *p* < 0.01. Full Western Blot Appendix A and Appendix A are provided in the Appendix A.

**Figure 3 animals-12-01477-f003:**
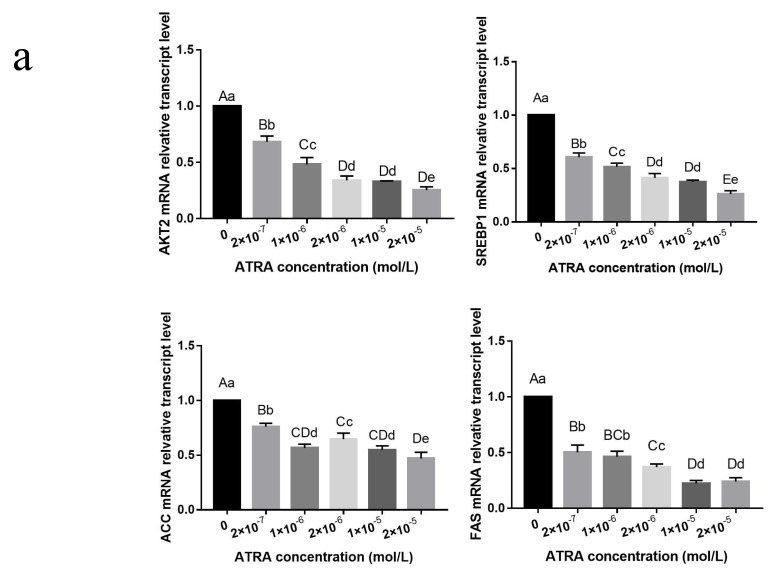
ATRA regulated the differentiation of adipocytes from Yanbian yellow cattle through the mTOR signaling pathway. (**a**) The expression levels of AKT2, SREBP1, ACC, and FAS mRNA were evaluated by qRT-PCR after treatment with different concentrations of ATRA. (**b**) The expression levels of SREBP1 and AKT2 proteins in bovine adipocytes treated with ATRA. Different lowercase letters in the figure indicate significant differences compared with the control group, *p* < 0.05. Different capital letters indicate highly significant differences compared with the control group, *p* < 0.01. Full Western Blot Appendix A and Appendix A are provided in the Appendix A.

**Figure 4 animals-12-01477-f004:**
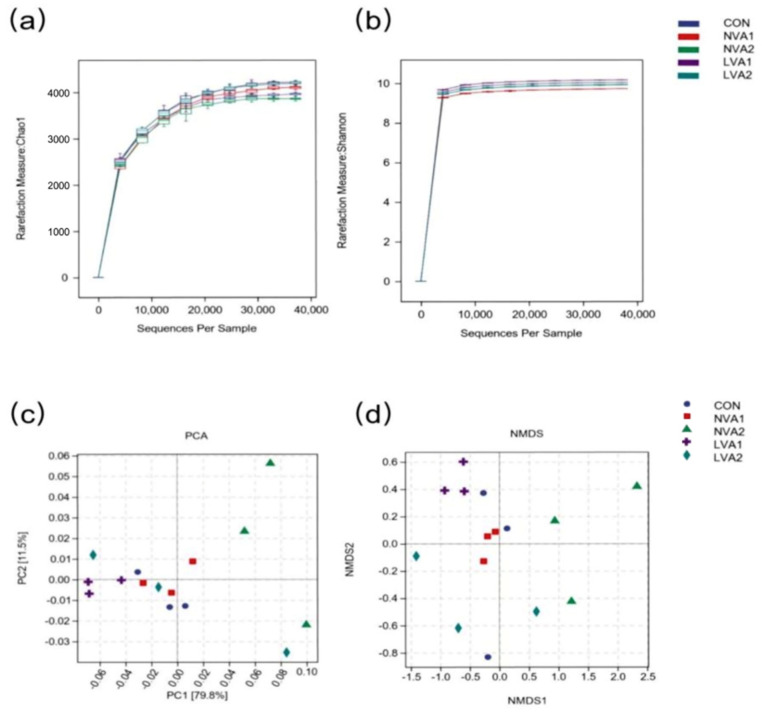
Changes in the intestinal flora diversity of Yanbian yellow cattle after vitamin A feeding. The α diversity analysis was carried out, including Chao1 richness estimator (**a**) and Shannon-wiener diversity index (**b**). β diversity analysis, including principal component analysis (PC) (**c**) and non-metric multidimensional scaling (NMDS)(**d**).

**Figure 5 animals-12-01477-f005:**
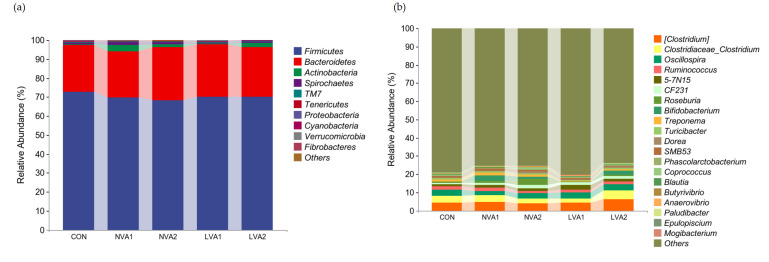
Effect of ATRA on the composition of intestinal microflora. The relative abundance of intestinal flora is shown in the stacked bar chart, including the change in taxa with significant differences at the phylum (**a**) and genus (**b**) levels.

**Table 1 animals-12-01477-t001:** Primer sequence of quantitative polymerase chain reaction used in this paper.

Primer Name	GenBank Accession No.	Primer Sequence (5′-3′)
Cyclin D1-F	NM_001046273.2	GCGTACCCTGACACCAATCTC
Cyclin D1-R		CTCCTCTTCGCACTTCTGCTC
CDK 2-F	NM_001014934.1	GGGTCCCTGTTCGTACTTATAC
CDK 2-R		CCACTGCTGTGGAGTAGTATTT
PCNA-F	NM_001034494.1	CCTTGGTGCAGCTAACCCTT
PCNA-R		TTGGACATGCTGGTGAGGTT
β-actin-F	NM_173979.3	AGGCATCCTGACCCTCAAGTA
β-actin-R		GCTCGTTGTAGAAGGTGTGGT
ERK 2-F	NM_175793.2	AAGACGCAACACCTCAGCA
ERK 2-R		AAGACGCAACACCTCAGCA
PPAR γ-F	NM_181024.2	CGAGAAGGAGAAGCTGTTGG
PPAR γ-R		TCAGCGGGAAGGACTTTATG
C/EBP α-F	NM_176784.2	TGGACAAGAACAGCAACGAG
C/EBP α-R		TCACTGGTCAACTCCAGCAC
ADPN-F	NM_015462053.1	AGGCAGAAAGGGAGAACC
ADPN-R		GTCGTGGTGAAGAGCAG
FAS-F	NM_001012669	AGGACCTCGTGAAGGCTGTGA
FAS-R		CCAAGGTCTGAAAGCGAGCTG
ACC-F	AJ132890	CATCTTGTCCGAAACGTCGAT
ACC-R		CCCTTCGAACATACACCTCCA
SREBP 1-F	NM_001113302	CTTGGAGCGAGCACTGAATT
SREBP 1-R		GGGCATCTGAGAACTCCTTGTC
AKT 2-F	NM_001206146.2	GCCGAATAGGAGAACTGGGG
AKT 2-R		CACGTCTGAGGTCGACACAA

**Table 2 animals-12-01477-t002:** Composition and nutritional level of total mixed rations (TMR) diet (dry matter basis).

Raw Materials	Content (%)	Nutritional Ingredient	Content (%)
corn	35.52	dry matter	87.63
soybean meal	14.40	crude protein	13.70
wheat bran	6.00	crude fat	3.20
straw	40.00	crude ash	10.69
salt	0.48	crude fibre	11.81
sodium bicarbonate	0.60	calcium	0.96
premix	3.00	phosphorus	0.62
total	100	Nacl	0.50

Notes: Per Kg premix contains: D-Biotin 2 mg; VD3 80KIU; VE 200 mg; VK 10 mg; Cu 400 mg; Fe 2000 mg; Mn 2000 mg; Zn 2000 mg; Se 10 mg; I 100 mg; Co 20 mg; Ethoxyquin 500 mg.

## Data Availability

The data are available by sending an email to the corresponding author.

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
