# Peer review of "Effects of Vitamin A on Yanbian Yellow Cattle and Their Preadipocytes by Activating AKT/mTOR Signaling Pathway and Intestinal Microflora"

_animals, 2022, doi:10.3390/ani12121477_

Round 1
Reviewer 1 Report
The present manuscript should be the study of the molecular mechanism of intramuscular fat deposition for ruminant animals. The manuscript description specialized in Yanbian yellow cattle is not suitable. In the case of present study the Yanbian yellow cattle should be one of the experimental animals of ruminants.
Simple summary should be written again as mentioned above.
Line14, the description of “As the earliest discovered vitamin,” should be deleted. Line14-17, the description of vitamin A that plays an important role in the fat accumulation in beef cattle, the reference should be indicated.
Line45-46, the description does not suitable.
Line53, describe the “coarse feeding tolerance” and “unique taste”.
Line73-84, in the present study the authors referenced both of monogastric animal and ruminant animal. Is it suitable? And the purpose of the present study is not clear.
Line161-174, the experiments design not clear including that of experiment period and treatment. And no description was found of the quantities of the feed.
Line386-391, the authors indicate the differences in fat deposition among individual animals, does it mean that Yanbian yellow cattle are special than other cattle?
Author Response
Point 1:The present manuscript should be the study of the molecular mechanism of intramuscular fat deposition for ruminant animals. The manuscript description specialized in Yanbian yellow cattle is not suitable. In the case of present study the Yanbian yellow cattle should be one of the experimental animals of ruminants.
Simple summary should be written again as mentioned above.
Response 1: Thank you very much for your valuable comments. I have revised the brief abstract.
Vitamin A is a fat-soluble vitamin that plays a role not only in vision and growth and development, but also in fat production and metabolism in animals. In order to improve the production of high-grade beef, it is necessary to explore the molecular mechanism of intramuscular fat deposition in beef cattle by molecular biology techniques. In this study, Yanbian Yellow cattle, one of the five major local breeds in China, were selected to investigate the effects of vitamin A and its metabolite all-trans retinoic acid (ATRA) on proliferation and differentiation of preadipocytes in Yanbian Yellow cattle, as well as the changes of intestinal microorganisms in Yanbian Yellow cattle after feeding vitamin A. It was found that ATRA inhibited adipogenic differentiation of preadipocytes in Yanbian Yellow cattle through AKT/mTOR signaling pathway. This study provides insight into nutritional management and suggests a role for vitamin A in lipid metabolism in Yanbian Yellow cattle.
Point 2:Line14, the description of “As the earliest discovered vitamin,” should be deleted.
Response 2: “As the earliest discovered vitamin,”has been deleted.
Point 3:Line14-17, the description of vitamin A that plays an important role in the fat accumulation in beef cattle, the reference should be indicated.
Response 3:In my search of this journal I found examples of references not being marked out in the Abstract section or Simple Summary section. Therefore I have added this sentence in the manuscript and marked the references.“This demonstrates that vitamin A plays an important role in fat accumulation in beef cattle and that its metabolite all-trans retinoic acid (ATRA) is considered to be a regulator of early adipocyte differentiation and lipid metabolism.” In addition the papers cited in this paper in which vitamin A plays an important role in fat accumulation in beef cattle are the following:9、19、28.
Point 4:Line45-46, the description does not suitable.
Response 4:Lines 45-46 of the revised manuscript: after successful revision, it is on lines 45-47.Vitamin A, also known as retinoic acid, is synthesized starting from β-carotene in the small intestine and liver of animals. Ruminants typically obtain vitamin A from plant sources and feed additives.
Response 5:Lines 53 of the revised manuscript: after successful revision, it is on lines 57.coarse feeding tolerance: It refers to that the animals are mainly fed with roughage, do not picky food, do not need to concentrate feed feeding, and are relatively easy to feed. Everything like grass, straw, and so on is roughage. Roughage is relative to concentrate feed. Concentrated feed is artificial feed.The more tolerant the animal is to roughage, meaning better tolerance to low protein and low energy feeds, the lower the feeding costs will be for the farmer and the more willing and easier to feed.The description of roughage tolerance in the paper is as follows: "Tolerance of roughage is actually a comprehensive index, the increase of crude fiber level in the ration inevitably leads to the decrease of crude protein and energy, thus the roughage tolerance of pigs also reflects the tolerance of pigs to low protein and low energy feeds to some extent." (Zhu M; Liu W; Zhang G. Research on roughage tolerance of local black pigs in Luci.Feed Research2012,07:64-66.doi:10.13557/j.cnki.issn1002-2813.2012.07.022.)
Point 6:Line73-84, in the present study the authors referenced both of monogastric animal and ruminant animal. Is it suitable? And the purpose of the present study is not clear.
Response 6:In fact, the research on the gut microorganisms of beef cattle is not well understood. Most of the studies on gut microorganisms are based on rodent models and are centered on the pathogenesis of human diseases and their potential mechanisms. Most of the research on gut microbes is based on rodent models and is centered on the pathogenesis of human diseases and their potential mechanisms, with little focus on whether they can influence lipid metabolism in livestock and poultry. Therefore, it gives us inspiration and reference for our experiment to investigate the role of gut microbes in lipid metabolism in beef cattle. However, based on the suggestions you gave, we amended the presentation in the manuscript to make a stronger logic.
Point 7:Line161-174, the experiments design not clear including that of experiment period and treatment. And no description was found of the quantities of the feed.
Response 7:In order to make the expression of experiment period, treatment and feed quantity clearer,we changed it to the following expression.Lines 161-174 of the revised manuscript: after successful revision, it is on lines 169-182.Detailed expressions of feed quantities can be found in Table 2 of the manuscript.
Before the start of the trial, we fed the cattle for 15 days according to the trial design as an adaptation period. We started the formal test after there were no anomalies. The formal trial diet was supplemented with 2,200 IU vitamin A/Kg dry matter (DM), the control group, according to the recommendations for beef cattle in the growth and fattening stages in the National Research Council publication "Nutritional Requirements of Beef Cattle (Eighth Revision)". The remaining twelve cattle were divided into the following experimental groups: the NVA1 group, fed 0 IU of vitamin A/Kg DM for 180 days; the NVA2 group, fed 0 IU of vitamin A/Kg DM for 240 days; the LVA1 group, fed 1,100 IU of vitamin A/Kg DM for 180 days; and the LVA2 group fed 1,100 IU vitamin A/Kg DM for 240 days. Vitamin A was purchased from DSM Vitamin Co., Ltd. (Changchun, China). During the study, all cattle were fed a standard basic diet and were free to drink water. The concentrate feeding was controlled at 1% of the body weight of each cattle and adjusted once a month according to the body weight of the cattle.The composition and nutritional components of the basic diet are presented in Table 2.
Point 8:Line386-391, the authors indicate the differences in fat deposition among individual animals, does it mean that Yanbian yellow cattle are special than other cattle?
Response 8:Thank you very much for noticing this.As one of the five local fine breeds in China, Yanbian Yellow Cattle has excellent meat quality and is a rare breed of beef cattle with Chinese characteristics. The practice of high-grade beef production in China has proved that Yanbian Yellow cattle have the genetic basis for the production of high-quality high-grade beef(Wang Z.; Liu H.; Yang Y. History, Development, Problems and Prospects of Yanbian Yellow Cattle. Agricultural Economics and Management 2014,(03):67-71. (in Chinese)), especially the ability to deposit fat in the muscle, marbling and other meat traits outstanding performance, but due to the large differences in fat deposition between individuals, directly affecting the grade of high-grade beef meat quality, resulting in high-grade beef meat quality grade rate of only 62%(Dong Y.; SUN X.;JIN Y.et al. Yellow cattle industry in Yanbian,Northeast China's Jilin province:Development and prospects.Meat Research 2014,6:30-33.(in Chinese)). Although the nutritional control method has increased the content of intramuscular fat to a certain extent, it has also increased the fat deposition in other parts such as viscera. Therefore, understanding the molecular mechanism of intramuscular fat deposition is particularly important for breeding practice of meat quality traits, and it has important certain guiding significance to improve the intramuscular fat deposition ability and beef quality of Yanbian yellow cattle.

Reviewer 2 Report
Dear authors
Many thanks for this valuable piece of work. I found meritorious of being published and I believe your results actually advance the present knowledge.
I enjoyed the structure and clarity of your work.
I have one only point which in my opinion may help to improve the overall background of the paper. I would suggest, if authors agree, to report that indeed we speak of vitamin A, but that for ruminants (and herbivores in general, with the exception of the suckling period of mother's milk, indeed in nature intake the precursor of vitamin A, which is beta- carotene. The different extent of beta-carotene conversion, relies at intestinal level, where the climate into a smaller molecule (one part is the vitamin A) allows absorption and distribution to reach the liver. Now, we know that essentially intensiey raised cattle have an intake of synthetic vitamin A and thus also the biological role may be different if compared to the natural form. The different extents however of the beta carotene conversion are species-specific and not all.carotenoids can be precursor to vitamin A of course. I would invite authors also to report some information about this and may refer to Cappai et al, 2017, Ecology and Evolution 7, 390-397. doi: 10.1002/ece3.2613
Congrats! Good job.
Author Response
Point 1:Dear authors
Many thanks for this valuable piece of work. I found meritorious of being published and I believe your results actually advance the present knowledge.
I enjoyed the structure and clarity of your work.
I have one only point which in my opinion may help to improve the overall background of the paper. I would suggest, if authors agree, to report that indeed we speak of vitamin A, but that for ruminants (and herbivores in general, with the exception of the suckling period of mother's milk, indeed in nature intake the precursor of vitamin A, which is beta- carotene. The different extent of beta-carotene conversion, relies at intestinal level, where the climate into a smaller molecule (one part is the vitamin A) allows absorption and distribution to reach the liver. Now, we know that essentially intensiey raised cattle have an intake of synthetic vitamin A and thus also the biological role may be different if compared to the natural form. The different extents however of the beta carotene conversion are species-specific and not all.carotenoids can be precursor to vitamin A of course. I would invite authors also to report some information about this and may refer to Cappai et al, 2017, Ecology and Evolution 7, 390-397. doi: 10.1002/ece3.2613
Congrats! Good job.
Response 1:
Dear Reviewers:
First of all, thank you very much for your advice, it is very important to me.
Secondly, I read the article that you recommended in your review comments. I think it is a good article. The discussion in the article addressing the synthesis of β-carotene, a precursor substance of vitamin A, in the small intestine and liver, and its involvement in visual function after light exposure along with other biological activities provides background support and complements the manuscript. New ideas are provided in response to photoperiodic changes in serum retinol and lack of supplementation of melanin in tissues in donkeys.
I am replying here to explain to you.
Thank you again for taking the time and effort to read my article and for your valuable comments.
Round 2
Reviewer 1 Report
The manuscript was well revised. Now it is suitable to publish in The Journal.
Author Response
Thank you very much for taking the time and effort to review our article again. Your review has made our article more professional and convincing. Once again, thank you very much for your comments.